# Yeast Bioflavoring in Beer: Complexity Decoded and Built up Again

Chiara Nasuti and Lisa Solieri *

Department of Life Sciences, University of Modena and Reggio Emilia, Via Amendola 2,
42122 Reggio Emilia, Italy; chiara.nasuti@unimore.it
* Correspondence: lisa.solieri@unimore.it; Tel.: +39-0522-522626

**Abstract:** Yeast is a powerful bioflavoring platform, suitable to confer special character and complexity to beer aroma. Enhancing yeast bioflavoring represents a chance for the brewing production chain to diversify its product portfolio and to increase environmental sustainability in the era of climate change. In flavor compound metabolism, multiple genes encoding biosynthetic enzymes and the related regulatory factors are still poorly known, but significant advances have been recently made to dissect gene contribution in flavor molecule production. Furthermore, causative mutations responsible for the huge strain diversity in yeast bioflavoring aptitude have been recently disclosed. This review covers the most recent advances in the genetics of yeast bioflavoring, with special regards to higher alcohols, esters, monoterpene alcohols, thiols, and phenolic derivatives of hydroxycinnamic acids. We also critically discussed the most significant strategies to enhance yeast bioflavoring, including bioprospecting for novel *Saccharomyces* and non-*Saccharomyces* strains, whole-genome engineering, and metabolic engineering.

**Keywords:** brewing yeasts; beer aroma; esters; monoterpene alcohols; thiols; higher alcohols; hydroxycinnamic derivatives; bioflavoring; nonconventional yeasts; metabolic engineering

## 1. Introduction

*Saccharomyces cerevisiae* is recognized as the workhouse in the production of alcoholic beverages worldwide. Beyond ethanol and $CO_2$, during alcoholic fermentation yeasts release several flavor-active compounds which confer peculiar aroma profiles to alcoholic beverages, such as higher/fuel alcohols, acetate/ethyl esters, carbonyls (aldehydes and ketones), and vicinal diketones (including unpleasant compounds such as diacetyl and pentanedione) (Table 1). In brewing, multiple factors differently contribute to the final organoleptic profile depending upon the beer style. In addition to hops, yeast fermentation is regarded as the most critical component affecting beer quality in almost all the brewing practices. In addition to the correct management of fermentation parameters, the choice of yeast strain is pivotal to determine the profile of aroma-active compounds. Therefore, rational selection and design of yeast cultures assures the production of beers with diverse organoleptic attributes. Based on these considerations, each type of beer has its own prevailing aroma profiles. For example, lagers, the most popular beer style worldwide, are produced at low temperature by bottom-fermenting strains currently assigned to *S. pastorianus* (interspecific hybrids between *S. cerevisiae* and *Saccharomyces eubayanus*) [1,2]. Generally, in lager beers isoamyl acetate (banana-like aroma) is the only yeast-derived metabolite with concentrations above the threshold level other than $CO_2$ and ethanol. In contrast, ale beers are produced by different types of *S. cerevisiae* top-fermenting strains [3,4] and have ethyl acetate (solvent-like aroma) and ethyl hexanoate (apple-like aroma) as additional flavoring compounds with levels above the threshold [5,6].

**Table 1.** Main flavor-active compounds produced or biotransformed by yeast during beer fermentation. The list includes fermentative metabolites and nonfermentative metabolites. Aroma thresholds and attributes were from Meilgaard [7], Haslbeck et al. [8], and Waterhouse et al. [9], except for thiols which were from Michel et al. [10].

| Compound Class | Compound (IUPAC Name) | Aroma Attributes | Aroma Threshold |
|---|---|---|---|
| Acetate esters | Ethyl acetate | Fruity, solvent | 30 mg/L |
| | Isoamyl acetate | Banana, apple, solvent | 1.2–2 mg/L |
| | 2-Phenylethyl acetate | Roses, honey, apple, sweetish | 0.2–3.8 mg/L |
| MCFA [1] ethyl esters | Ethyl butyrate | Papaya, butter, sweetish, apple | 0.4 mg/L |
| | Ethyl decanoate | Fatty acids, apple, solvent | 1.5 mg/L |
| | Ethyl hexanoate | Apple, fruity | 0.2–0.23 mg/L |
| | Ethyl octanoate | Apple, aniseed | 0.9–1.0 mg/L |
| Higher alcohols | n-Propanol (Propan-1-ol) | Alcohol | 600 mg/L |
| | isobutyl alcohol (2-methylpropan-1-ol) | Solvent | 100 mg/L |
| | Isoamyl alcohol (3-Methylbutan-1-ol) | Alcohol, banana | 50–65 mg/L |
| | 2-Methyl-1-butanol or active amyl alcohol (2-Methylbutan-1-ol) | Alcohol, solvent | 50–70 mg/L |
| | 2-Phenylethanol | Floral, rose | 40 mg/L |
| Monoterpene alcohols | Geraniol | Floral, roses, fruity, citrus | 6 µg/L |
| | Nerol | Lemon, fruity | 500 µg/L |
| | Linallol | Floral, citrus | 3–5 µg/L |
| | β-Citronellol | Floral, fruity, citrus | 8 µg/L |
| Thiols and their esters | 3-mercaptohexan-1-ol | Passion fruit, grape fruit, gooseberry, guava | 55–60 ng/L |
| | 3-mercaptohexyl acetate | Passion fruit, grape fruit, box tree, gooseberry, guava | 4 ng/L |
| | 4-mercapto-4-methylpentan-2-one | Box tree, passion fruit, broom, black currant | 1.5 ng/L |

[1] MCFA, medium-chain fatty acids.

While several flavor-active compounds are secondary metabolites produced by yeasts during alcoholic fermentation, yeasts can also enzymatically convert odorless precursors originating from hops and malt into flavor-active compounds, an activity called biotransformation [11]. The most studied biotransformation reactions entail the release of polyfunctional thiols and terpene alcohols catalyzed by β-lyase and β-glucosidase enzymes, respectively. These reactions can have an important impact on the economic and environmental sustainability of brewing process as hops and malt represent important raw material costs in the brewing industry, and their cultivation requires high demand for water, land, and other agricultural resources [12,13]. Improving the yeast biotransformation of hop-derived neutral precursors into flavor-active molecules could reduce the usage of hops in brewing, as well as provide beers with diverse organoleptic profiles [14].

New trends in product differentiation and diversification are emerging in several food sectors, including the brewing industry [15]. Despite the dominance of lagers across the world, the so-called craft beers have increasingly transformed the global beer market in the past few decades, favoring specialty hopped beers produced by local breweries and stopping the homogenization of beer by the global multinational producers [16]. This demand has driven the search for novel yeast cultures targeting specialty beers, and two main cutting-edge topics are on the forefront of brewing innovation. The first topic deals with the exploitation of yeast diversity to expand the genetic portfolio of brewing starters and the resulting final organoleptic profile, as previously reviewed [17–20]. The second topic deals with the dissection of genetic determinants responsible for the huge strain diversity in the release of flavor-active compounds [21,22]. There is a strict interplay between these research topics. The improved knowledge of genes responsible for flavor-

active phenotypes can be useful in the rational design of novel brewing cultures, as well as in marker-assisted selection of new "wild" *Saccharomyces* and non-*Saccharomyces* isolates to introduce in the brewing process.

While several reviews described how hops impart characteristic flavor and aroma to the beer and improve the microbiological stability of the finished product [23], other works described the contribution of yeasts to beer aroma and flavor, focusing either on desirable odorants produced during fermentation or on biotransformation reactions of hop and malt precursors [12,24–27]. This paper will include both categories of yeast-derived flavor-active molecules, limiting the discussion of the most recent insights on the metabolic pathways and genes involved in their production. Furthermore, we will provide an overview of the most recent attempts in bioprospecting and metabolic engineering to enhance and diversify the bioflavoring contributions of brewing yeasts.

## 2. Flavor-Active Compounds Derived from Yeast Metabolism

### 2.1. Higher Alcohols

Higher alcohols, often referred to as "fusel alcohols", are alcohols with more than two carbon atoms which contribute both positively and negatively to the beer bouquet [28]. These compounds include three distinct classes derived from branched-chain, sulfur-containing, and aromatic amino acids, respectively. Amounts of higher alcohols higher than 300 mg/L can lead to a pungent smell and taste in beer, whereas optimal levels impart desirable characteristics. The optimum concentration of higher alcohols in 12.0 °P beers brewed via bottom fermentation is 70–120 mg/L [29]. Isoamyl alcohol (also known as 3-methylbutan-1-ol) is an important yeast-derived compound for beer flavor. If isoamyl alcohol increases too much, beer is poorly drinkable and is perceived as tasting heavier [30]. Isobutyl alcohol becomes undesirable when the concentration surpasses 20% of the total amount of isobutyl alcohol, n-propanol, and isoamyl alcohol [31]. The higher alcohol 2-phenylethanol, associated with a rose flavor, is typically below the sensory threshold in beer, but it may still be detected by additive interactions with isoamyl and isobutyl alcohol if these alcohols are present at high concentrations [5]. Other than for their intrinsic aromatic attributes, higher alcohols are significant in determining beer quality since they are strongly related to esters formation [32].

Higher alcohols are produced during fermentation, especially under anaerobic conditions, through the decarboxylation of α-keto acids to the corresponding aldehydes and the reduction of aldehydes to alcohols (Figure 1) [33]. Therefore, higher alcohols contribute to the redox balance inside the cell and are less toxic than the fusel acid counterparts. The α-keto acids precursors are derived from two pathways related to amino acid and sugar metabolisms, respectively: (1) the catabolic consumption of amino acids assimilated from the medium via the Ehrlich pathway; (2) and de novo amino acid synthesis, also known as the anabolic pathway [26] (Figure 1). By examining the origin of fusel oils in wine, Hazelwood et al. (2012) concluded that they are derived from amino acids produced internally, rather than from the breakdown of amino acids found in the grape juice [33]. Additional evidence supported that the synthesis of n-propanol, a specific type of fusel oil, exclusively occurs through the anabolic pathway in wine [34,35]. In brewing, free amino acid nitrogen (FAN) is generally high, and brewer's yeasts mainly produce higher alcohols through the Ehrlich pathway [25,33].

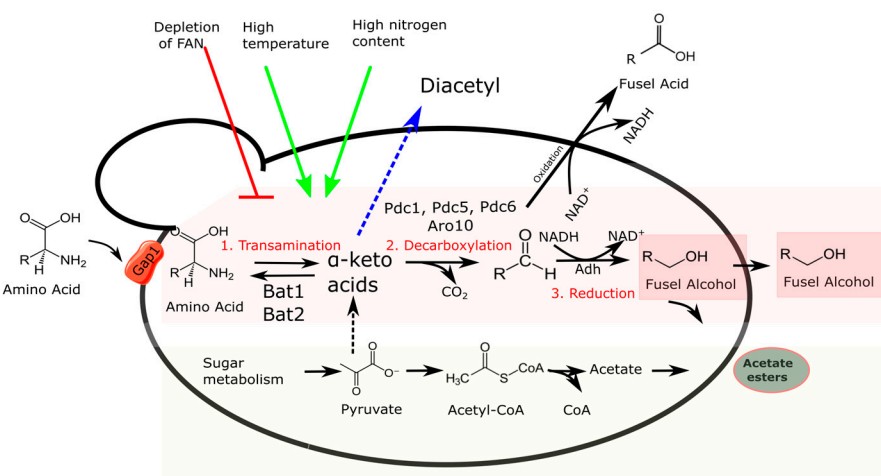

**Figure 1.** Production of higher alcohols during wort fermentation. Both the catabolic and anabolic pathways are summarized. In the Ehrlich pathway, amino acids are taken in through the Gap1 transporter and undergo transamination via amino acid aminotransferases, resulting in α-keto acids. α-keto acid dehydrogenases reduce α-keto acids into aldehydes which are then further converted into either alcohols or acids. In the anabolic pathway, the breakdown of sugars produces the main precursors of α-keto acids such as pyruvate; α-keto acids are then used for amino acid biosynthesis. Only the most important genes are reported for brevity. Dotted black arrows indicate omitted reactions, while dotted blue arrows indicate chemical reactions. Green and red arrows indicate the positive and negative factors affecting higher alcohol production, respectively. FAN, free amino acid nitrogen; Bat1, mitochondrial branched-chain amino acid aminotransferase; Bat2, cytoplasmic branched-chain amino acid aminotransferase; Pdc1, pyruvate decarboxylase isozyme 1; Pdc5, pyruvate decarboxylase isozyme 5; Pdc6, pyruvate decarboxylase isozyme 6; Aro10, Phenylpyruvate decarboxylase; Adh, alcohol dehydrogenase; Gap1, General amino acid permease.

In the Ehrlich pathway, amino acids are taken from the surrounding medium inside the cell and submitted to transamination via amino acid aminotransferases found in both the mitochondria and cytosol [eden34]. These enzymes transform amino acids into α-keto acids, which are then reduced into aldehydes via α-keto acid dehydrogenases. The aldehydes are then further converted into either alcohols or acids, depending on whether they undergo reduction or oxidation reactions, respectively (Figure 1). Production of higher alcohols involves more than 150 genes, as recently revised [25]. The most studied genes are listed in Table 2. For example, the branched-chain amino acids valine (precursor of isobutyl alcohol), leucine (precursor of isoamyl alcohol), and isoleucine (precursor of active amyl alcohol) are converted into the corresponding α-keto acids α-ketoisovalerate, α-keto-β-methylvalerate, and α-ketoisocaproate via mitochondrial and cytoplasmic aminotransferases Bat1 and Bat2, respectively. Bat2 is mainly involved in the production of higher alcohols, while Bat1 is mainly involved in amino acid biosynthesis. Transaminases Aro8 and Aro9 mainly act on phenylalanine, tryptophan, and tyrosine. Many studies have investigated the effects of *BAT1* and *BAT2* deletion or overexpression on higher alcohol content in beer, as previously reviewed [25,32,33]. More recently, Bat1 and Bat2 enzymatic variants were proven to increase higher alcohol production, suggesting that engineering aminotransferases via specific amino acid substitutions could be a useful strategy to modulate fusel alcohol content in beer [36]. In the yeast genome, five genes encode enzymes catalyzing the decarboxylation of 2-oxo-acids derived from amino acid transamination, namely pyruvate decarboxylase genes (*PDC1, 5*, and *6*), *ARO10*, and *THI3* [35]. Aro10 is proposed as the major decarboxylase involved in phenylalanine and tryptophan catabolism. The *S. pastorianus* aneuploid genome harbors four copies of *ARO10*, three from *S. cerevisiae* (*LgScerARO10*) and one from *S. eubayanus* (*LgSeubARO10*). Bolat et al. [37] proved that these orthologs undergo subfunctionalization since LgScAro10 orthologs are mainly involved in

higher alcohol production in the catabolic Ehrlich pathway, while LgSeubAro10 produces higher alcohols in the anabolic pathway.

**Table 2.** Main genes responsible for producing higher alcohols in yeasts.

| Gene | Functions | References |
|---|---|---|
| *TIR1* | Cell wall mannoprotein for maintaining the integrity of the cell wall | [38,39] |
| *GAP1* | Amino acid transporter | [38] |
| *BAT1* | Transaminase which transfers the amino groups between branched-chain amino acids and α-keto acids | [40,41] |
| *BAT2* | Transaminase which transfers the amino groups between branched-chain amino acids and α-keto acids | [40,41] |
| *PDC1* | Decarboxylase which removes the carboxyl group from an α-keto acid | [42] |
| *PDC5* | Decarboxylase which removes the carboxyl group from an α-keto acid | [43] |
| *ARO10* | Decarboxylase which removes the carboxyl group from an α-keto acid | [44] |
| *ADH1* | Alcohol dehydrogenase which reduces aldehyde to alcohol using NADH as a cofactor | [45,46] |
| *ARO4* | 3-deoxy-7-phosphoheptulonate synthase which synthesizes a tyrosine precursor | [47] |

In anabolic metabolism, the production of α-keto acids is no longer dependent on transamination reactions. Instead, it is dependent on the breakdown of sugars and pyruvate functions as the main precursor of α-keto acids [26] (Figure 1). The most important anabolic pathway is the isoleucine–leucine–valine (ILV) pathway, responsible for the de novo synthesis of branched-chain amino acids from sugar catabolism. The ILV pathway comprises acetolactate synthase (ALS, encoded by *ILV2*), ketol-acid reductoisomerase (KARI, encoded by *ILV5*), and dehydroxyacid dehydratase (DADH, encoded by *ILV3*) and has α-acetolactate and α-hydroxybutyrate as the main intermediates [47]. Spontaneous oxidative decarboxylation of α-acetolactate and α-hydroxybutyrate can occur outside the cell, releasing diacetyl and 2,3-pentanedione, respectively. These molecules are vicinal diketones and are considered serious off-flavors, especially in lager beer, for their distinct buttery/butterscotch flavor [48]. Towards the end of fermentation, diacetyl can be reabsorbed by the cell and converted to acetoin (and subsequently 2,3-butanediol) via various reductases called Old Yellow Enzymes (Oye) [49]. Recently, diacetyl was reduced by 17.83% via *ILV2* gene deletion and by 28.26% via the concomitant *ILV2* gene deletion and *BDH1* gene overexpression [50]. *BDH1* encodes butanediol dehydrogenase, which catalyzes the conversion of acetoin to 2,3-butanediol which has a higher perception threshold than diacetyl.

Several extrinsic factors affect the production of fusel oils by yeasts, including wort aeration, temperature, and wort composition [48,51–53] (Figure 1). Kucharczyk and Tuszyński (2017) tested the effects of different wort aeration regimes on the flavor compounds profile and proved that with the increase in wort aeration, the concentration of higher alcohols increased without any effect on esters [54]. Generally, high temperatures positively contribute to higher alcohols, while low temperatures inhibit nitrogen metabolism and, consequently, higher alcohols production [42–51] (Figure 1). FAN is one of the most relevant compositional parameters affecting higher alcohol production [48,55]. High-gravity wort enriched in proteinaceous substances favors the production of higher alcohols. This explains why

top-fermented wheat beers contain high concentrations of higher alcohols which generate disagreeable flavors [56]. Amino acids are gradually taken up by yeast cells through channels and transporters, such as Gap1, Lyp1, and Mup1, which are situated in the cell membrane (Table 3) [57]. Among these, the general amino acid permease Gap1 catalyzes the active transport of all the D- and L-isomers of the amino acids [58]. Gap1 is an inducible amino acid permease that is regulated via amino acid abundance, a phenomenon called nitrogen catabolic repression (NCR). NCR assures the yeast capability to select preferred and more abundant nitrogen sources and prevents the uptake of poorer nitrogen [26]. Gap1 has been recently described as a core regulatory gene in the production of higher alcohols [38], and *GAP1* gene deletion is reported to decrease levels of diacetyl [59] and higher alcohols [60]. Furthermore, His supplementation inhibits the NCR-mediated downregulation of *GAP1* gene during high-gravity wort fermentation with a lager strain, favoring the release of higher alcohols and esters [61]. All this evidence links nitrogen utilization to higher alcohol production. Accordingly, Scott et al. (2021) studied the diversity of four wine strains in releasing flavor-active compounds during must fermentation and reported that total aroma production is a function of nitrogen utilization. In particular, the strain variability in leucine utilization explains why wine yeasts are so different in the release of higher alcohols and corresponding acetate esters [62].

**Table 3.** Primary amino acid transporters in yeast cells. These transporters are categorized based on their respective families and the corresponding biological functions are indicated.

| Transporter | Function | Family |
| --- | --- | --- |
| Gap1 | General AA transporter and nutrient sensor | |
| Agp1 | Transporter of all AAs except for arginine and lysine | |
| Agp2 | AA transporter and sensor that regulates the expression of importers for carnitine and polyamines | Plasma membrane transporter |
| Bap2 | Leucine transporter | |
| Gnp1 | High affinity glutamine, cysteine, and proline permease | |
| Put4 | Proline, alanine, and glycine transporter | |
| Lyp1 | Lysine transporter | |
| Alp1 | Arginine transporter | |
| Avt3 | General neutral AA vacuolar exporter | |
| Avt4 | General neutral AA vacuolar exporter | |
| Avt7 | Vacuolar exporter of glutamine and proline | Vacuolar transporter |
| Vba5 | Plasma membrane transporter for arginine and lysine | |
| Agc1 | Aspartate/glutamate carrier | Mitochondrial transporter |
| Ort1 | Ornithine, arginine, and lysine carrier | |
| Uga4 | Vacuolar membrane protein | Other transporter |

AA, amino acids.

### 2.2. Esters and Fatty Acids

Volatile esters are the most important yeast-derived contributors to the aroma of flowers and ripe fruits in wine and beer. Esters generally have a low odor threshold (Table 1) and minor changes in their concentrations dramatically impact beer quality [5]. Esters' contribution to beer aroma strongly depends on the ratio between esters and higher alcohols. In lager beer, a 3–4:1 ratio of higher alcohols to esters is acceptable, while a higher ratio results in a dry taste and a less aromatic characteristic of the beer [29].

During yeast fermentation, two main classes of esters are released, namely acetate esters and fatty acid ethyl esters (FAEEs). Acetate esters arise from condensation between

acetyl-CoA and ethanol or higher alcohols or any other alcohol derived from aldehyde reduction (Figure 2). In ethyl esters formation, ethanol represents the alcohol coreactant whereas either acetate or medium chain fatty acids (MCFA) generally act as an acyl moiety. Significant progress has been made in elucidating the biochemical pathways responsible for the synthesis of both classes of esters. However, our knowledge in this respect is not complete yet, especially for ethyl esters.

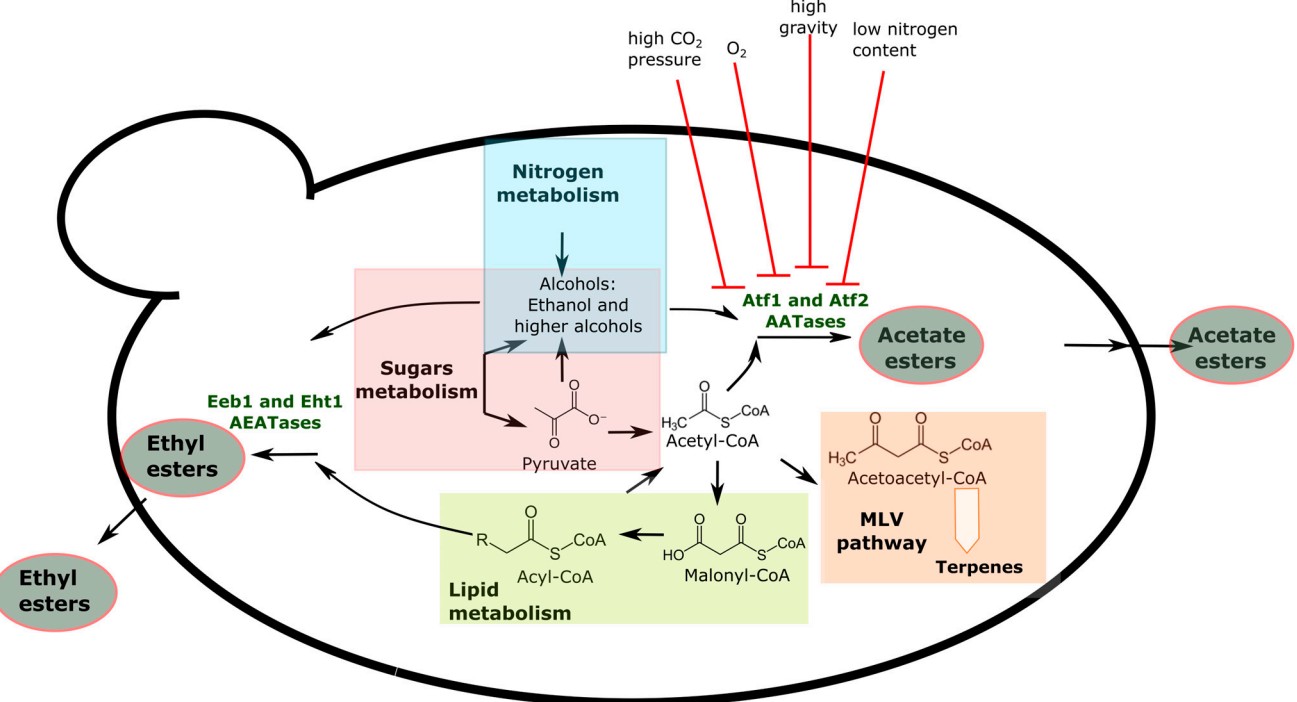

**Figure 2.** Overview of the main pathways for ester biosynthesis in yeast. Centrality of acetyl-CoA is underlined. Lipid biosynthesis and MVA pathways are omitted for brevity. Extrinsic factors negatively affecting *ATF1* and *ATF2* gene expression are reported in red. AATase, Alcohol acetyl transferases; AEATase, acyl-CoA:ethanol O-acyltransferases; MVA, mevalonate; Atf1, Alcohol acetyl-transferase 2; Atf2, Alcohol acetyltransferase 2; Eeb1, Acyl-coenzymeA:ethanol O-acyltransferase involved in ethyl ester biosynthesis (Eeb); Eht1, Ethanol Hexanoyl Transferase.

Alcohol acetyl transferases I and II (AATase I and II; EC 2.3.1.84), encoded by *ATF1* and *ATF2* genes, synthetize acetate esters [63,64]. Atf1 is responsible for the production of isoamyl acetate, which is found at concentrations of 0.8–3.8 mg/L in beer and confers a banana descriptor when it is above the aroma threshold [5] (Figure 2). *ATF1* gene expression is directly related to acetate esters production. In a laboratory yeast strain, *ATF1* deletion decreased the levels of isoamyl acetate more than 80%, the levels of phenyl ethyl acetate by 75%, and the levels of ethyl acetate by more than 50%, while constitutive overexpression of *ATF1* resulted in a 180-fold increase in isoamyl acetate concentration [65]. However, the production of acetate esters was not entirely abolished in a Δ*Atf1*Δ*Atf2* double mutant, suggesting that other genes are involved in acetate ester production other than *ATF1* and *ATF2*. Recently, YGR015C (currently named *EAT1*) and YGR031W (named *IMO32* by [66]) have been identified as involved in ethyl acetate production [67,68]. However, ester synthesis was not completely abolished in a *S. cerevisiae* strain lacking all AATase-encoding genes [69]. Extrinsic factors, such as $O_2$, high sugar concentration, nitrogen depletion [70,71], and, recently, high $CO_2$ pressure [72], have been described to negatively affect AATase-encoding gene transcription.

Among ethyl esters, MCFA ethyl esters such as ethyl butyrate, ethyl caproate, ethyl octanoate, and ethyl decanoate are formed via the condensation of ethanol and fatty acid acyl-CoA or fusel acyl-CoA via acyl-CoA:ethanol O-acyltransferases (AEATases) [64]. Fusel

acyl-CoA is a byproduct of carbon and nitrogen sources that predominantly originate from the degradation of α-keto acids, while fatty acid acyl-CoA is produced through the biosynthesis of fatty acids. The key enzymes involved in MCFA ethyl esters synthesis are Eht1 (ethanol hexanoyl transferase 1) and Eeb1 (ethyl ester biosynthesis 1), which are, respectively, encoded by the paralogs *EHT1* and *EEB1* (Figure 2) [73]. As the double-deletion strain BY4741 Δ*Eeb1*Δ*eht1* and the single-deletion strain BY4741 Δ*eeb1* are similar in levels of MCFA ethyl esters, Eeb1 is considered more important in MCFA ethyl ester synthesis than Eht1 [69]. Overexpression of *EHT1* and *EEB1* genes has conflicting effects based on strain background. Saerens et al. [73] reported that *EHT1* overexpression does not increase ethyl ester level due to the bifunctional role of Eht1 as AEATase and esterase, while Lilly et al. [74] documented a significant increase in ethyl esters in a wine yeast mutant over-expressing *EHT1*. As acyl-CoA enzymes are precursors for MCFA ethyl esters (Figure 2), genetic elements and extrinsic factors affecting lipid biosynthesis and homeostasis of lipid-related processes strongly impact ethyl esters.

Significantly, a mutant strain lacking all genes encoding both AATase and AEATase still exhibits esters production [69]. This evidence suggests that esters formation is so important for cell homeostasis that several genes have complementary and redundant functions in ester synthesis. Four main hypotheses have been formulated to explain the role of ester formation. In the first hypothesis, the absence of oxygen induces acetyl-CoA accumulation and depletion of free CoA. Ester formation could be a way to release free CoA under anaerobic conditions. Accordingly, $O_2$ inhibits ester synthesis (Figure 2). Secondly, it is well-known that under anaerobic conditions like those occurring during beer fermentation, complete depletion of $O_2$ inhibits unsaturated fatty acid synthesis, negatively impacting membrane fluidity and cell viability. Mason and Dufour [63] suggested that yeast cells take advantage of ester formation as certain esters with long-chain hydroxy fatty acids could serve as unsaturated fatty acid analogues. Thirdly, ethyl ester formation could be a detoxification system against MCFAs which are toxic for the cells, especially at a low pH. Esterification can positively impact intracellular pH homeostasis, as MCFA ethyl esters can diffuse more easily through the plasma membrane into the surrounding medium than the corresponding MCFA. Finally, esters are attractive for insects like *Drosophila*, favoring yeast dispersion and ecological success.

## 3. Flavor-Active Compounds Originated from Beer Precursors by Yeast Enzymes

In brewing, the volatile fraction of hops (*Humulus lupulus*), collectively known as "hop oil", is a very heterogeneous mixture of hundreds of compounds which account for 0.5–3% of hop dry matter and impart a whole range of aromas to beer, such as herbal, spice, floral, citrus, fruity, and pine aromas. The hop oil composition strongly varies depending upon hop variety, cultural conditions, storage, and processing parameters [75], and several attempts have been made to categorize aroma-active compounds of hop oils into main chemical groups (for a review see [76]). The mostly accepted classification was proposed by Sharpe and Laws [76], which identified three main groups of hop-associated aroma compounds, such as hydrocarbons (containing monoterpenes, sesquiterpenes, and aliphatic hydrocarbons, representing 50–80% of hop essential oil), oxygen-containing compounds (terpene alcohols, sesquiterpene alcohols, and other oxygenated compounds, accounting for 30% of hop essential oil), and sulfur compounds (thioesters, sulfides, and other sulfur compounds). The quantitatively most abundant terpenoids in hops are the sesquiterpenoids, such as α-humulene, β-caryophyllene, and β-farnesene, and the monoterpenoid myrcene.

Yeast can absorb hydrocarbons on the cell wall and indirectly reduce them during wort fermentation. In addition, yeast can be metabolically active on terpenoids. Here, we focused the discussion on yeast-derived biotransformation of monoterpene alcohols, a special group of terpenoids, which are primary flavor determinants in hopped beer.

### 3.1. Glucosidase Activity and Monoterpene Alcohols

Terpenoids derive from C5-building blocks such as isopentenyl pyrophosphate (IPP) and its isomer dimethylallyl pyrophosphate (DMAPP). *S. cerevisiae* has a well-known mevalonate (MVA) pathway which produces IPP and DMPP for the downstream biosynthesis of essential metabolites including hemes and sterols. The extent of de novo biosynthesis of monoterpene alcohols is generally irrelevant during wort fermentation as a monoterpene synthase encoding gene is missing in the yeast genome [77]. Takoi [78] reported that, independently from hop varieties, during wort fermentation, linalool and α-terpinol slightly decrease, while β-citronellol, nerol, and geraniol increase. Yeasts convert geraniol into linalool or β-citronellol, nerol into geraniol, and α-terpinol or linalool, and the latter into α-terpinol [79,80]. The NADPH oxidoreductase Oye2 containing flavin mononucleotide (FMN) is the main enzyme responsible for geraniol reduction into β-citronellol during wort fermentation [80] (Figure 3). Furthermore, yeasts can carry out acetylation of monoterpenes via the AATase enzyme Atf1 or can break down monoterpene esters via esterases or lipases [81]. King and Dickinson (2000, 2003) reported that monoterpene acetate esters are formed in high concentration in late-hopped beer via yeast metabolism and that lagers are more effective than ale strains in this metabolic reaction [79,80]. More recently, ale yeast strains have been proved to produce high amounts of citronellol acetate (floral, fruity, pear, and apple character) and nerol acetate (floral and green flavor) without any correlation between higher alcohol acetate esters and monoterpene acetate esters [82]. Yeasts were reported to also convert acetylate geraniol into geranyl acetate [82].

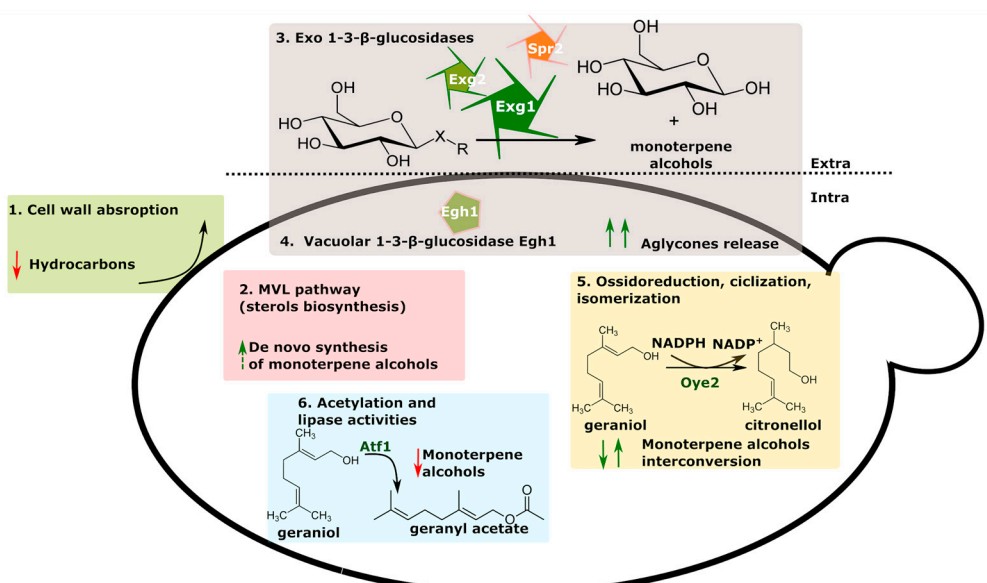

**Figure 3.** Yeast-catalyzed biotransformations of monoterpene alcohols during hopped wort fermentation. Red and green arrows indicate decreases and increases in the concentration of terpene alcohols, respectively. Dotted green arrows represent slight increases. For any class of biotransformation, only the main reactions and enzymes are depicted for clarity.

Yeast-derived β-glycosidase activity is responsible for the most relevant increase in monoterpene alcohols during wort fermentation (Figure 3). A relevant fraction of monoterpenoids is present in hopped wort as glycosidically conjugated flavor-inactive precursors (glycosides), which can release their aromatic counterparts, called aglycones, via the cleavage of sugar moieties. The hydrolysis of glycosides can be either acidic, as that which occurs in sour beer fermentation, or enzymatic. In *Saccharomyces* and non-*Saccharomyces* yeasts, β-glycosidases are responsible for the release of β-D-glucose and aglycones, such as linalool. Daenen et al. [83] firstly demonstrated that the *EXG1* gene encodes an extra-cellular 1,3-β-glucanase involved in cell wall integrity and responsible for monoterpene alcohol release. Exg1 was demonstrated to be active on a broad spectrum of

glycosides, such as flavonoid glucosides, flavones, flavonols, and isoflavones [84]. *EXG2* and *SPR1* are two additional extra-cellular 1,3-β-glucanase encoding gene paralogs of *EXG1*. Sharp et al. [75] demonstrated that lager and ale yeasts strongly differ in 1,3-β-glucanase activity.

### 3.2. β-Lyase Activity and Thiols

Hops are rich in valuable thiols that have very low perceiving thresholds (on the order of ng/L) and play a crucial role in the overall beer aroma. These compounds can be present in hops as odorless precursors, such as glutathione S-conjugate and cysteine S-conjugates. Among these, cysteinylated precursors (cys-X), such as Cys-4MMP (cysteinylated form of 4-mercapto-4-methylpentan-2-one), Cys-3MH (cysteinylated form of 3-mercaptohexan-1-ol), and Cys-3MHA (cysteinylated form of 3-mercaptohexylacetate), can be converted into the flavor-active forms via yeasts β-lyase activity [26,27,85]. This reaction ultimately releases pyruvate, ammonium, and a sulfur-containing fragment into the cell (Figure 4).

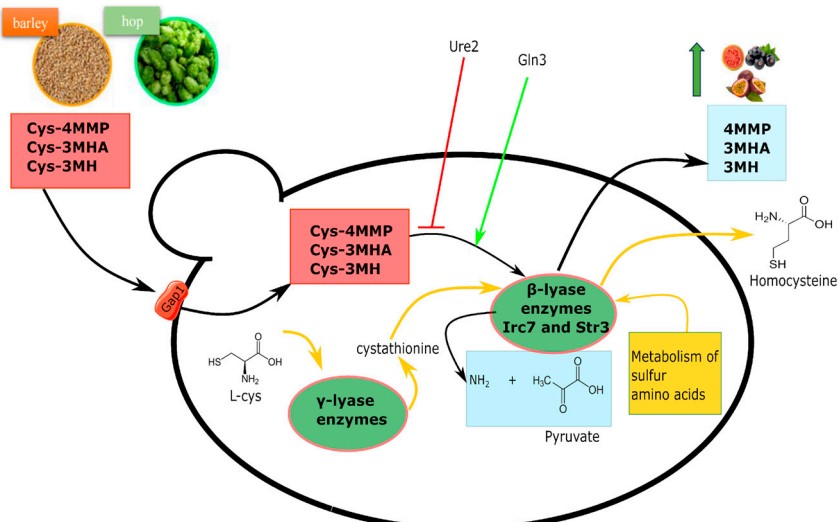

**Figure 4.** Graphical representation of the role of β-lyase enzymes in converting odorless cysteine conjugates (cys-X) present in barley and hops into fruity volatile thiol compounds. Gap 1 is an amino acid transporter, enabling the cysteinylated precursor to enter the cell. Once inside, the beta-lyase enzymes Irc7 and Str3 break down cys-X, liberating pyruvate, ammonia cysteine, and a thiol compound, which are then expelled from the cell. Green and red arrows indicate factors positively and negatively affecting β-lyase activity, respectively. Yellow arrows indicate reactions belonging to cysteine metabolism. 4MMP, methylpentan-2-one; 3MH 3-mercaptohexan-1-ol; 3MHA, 3-mercaptohexyl acetate.

The cell readily absorbs cys-X thanks to the presence of Gap1 [86], an amino acid transporter that specifically facilitates the entry of cysteine and other amino acids, as reported above. Once inside, cys-X is subject to direct cleavage by a group of enzymes encoded by *IRC7*, *STR3*, *BNA3*, *CYS3*, and *GLO1* genes [26,87]. These enzymes, known as cysteine S-conjugate β-lyases or cystathionine β-lyase (EC 4.4.1.8), use pyridoxal 5′-phosphate as a cofactor and are involved in methionine metabolism. Through a β-elimination reaction, they break down cysteine S-conjugates that bear an electron-withdrawing group attached to the sulfur [88]. Among them, Irc7, followed by Str3, was described as a key determinant of 4-MMP release from Cys-4-MMP and as a significant contributor of 3-MH release from Cys-3-MH [89–91] (Figure 4).

Brewing strains are highly variable in their thiol release ability [91]. Like higher alcohols, volatile thiols can be influenced by NCR, which involves the targeted inhibition of GATA transcriptional activators [92]. The NCR regulation of these genes may be strongly strain dependent [93]. Ura2 is an NCR transcriptional regulator and prevents the full ex-

pression of many genes under rich nitrogen sources. Specifically, the primary transcription factor responsible for thiols release, Gln3, is hindered when there is an excess of the Ure2 protein in nitrogen-rich environments. Previous work has demonstrated that, when a functional *IRC7* gene is present, *URA2* deletion is enough to increase the release of thiols [92,94]. Therefore, a strain defective in NCR should exhibit a strong thiols release phenotype.

Other than NCR regulation, allelic variability in the *IRC7* gene is responsible for the interstrain diversity in releasing thiols. The full-length gene, known as $IRC7^L$, encodes a 400-amino-acid long functional protein suitable to release thiols [89]. Conversely, in many strains, a short allelic variant (the so called $IRC7^S$) results from a premature stop codon which causes a 38-bp deletion. Consequently, the $IRC7^S$ variant encodes a truncated protein with only 340 amino acids, which is unable to assure β-lyase activity and to release aromatic thiols [26,90,91]. Given the potential presence of both heterozygous and homozygous yeast strains, three distinct genotypes can be found in brewing strains: individuals homozygous for the short allele ($IRC7^S$/$IRC7^S$), heterozygous individuals ($IRC7^S$/$IRC7^L$), and homozygous individuals for the full-length allele ($IRC7^L$/$IRC7^L$).

In addition to this 38-bp deletion, a considerable number of SNPs were found on the *IRC7* gene that affect the yeast strain's ability to release thiols from cysteinylated precursors [26]. Notably, one of the most common SNPs involves the replacement of threonine with alanine at position 185, causing the loss of a crucial hydrogen bond with Asp183's carboxylate group. This bond is essential for optimal interaction with the PLP cofactor's pyridine nitrogen. As a result, the β-lyase enzyme harboring this point mutation exhibits an activity reduction of about 50% compared to the wild-type enzyme [26].

### 3.3. Ferulic Acid Decarboxylase and Phenolic Off-Flavour Compounds

Phenylpropanoids are plant-derived compounds originated from phenylalanine and, to a lesser extent, tyrosine. Among these, hydroxycinnamic acids can undergo decarboxylation resulting in volatile phenols, which impart pleasant spicy, clove-like, sweet, and vanilla-like flavor notes to beer [95,96]. The most common of these are 4-vinylguaiacol (4VG), 4-vinylphenol, 4-ethylguaiacol, 4-ethylphenol, 4-vinylsyringol, styrene, eugenol, and vanillin [97]. In bottom-fermented lager and pilsner-type beers, volatile monophenols are considered as off-flavors (POF). However, volatile monophenols are essential for the typical aroma of top-fermented beers like German Weizen, Rauch beers, and Belgian white beers [98]. Differently from their acid counterparts, volatile monophenols are strongly flavor-active: for instance, 4VG derived from ferulic acid exhibits a flavor threshold below 20 ppb in water [99].

In addition to acidic decarboxylation, volatile phenols can be produced during yeast fermentation. According to their capability or incapability to decarboxylate hydroxycinnamic acids into phenols, brewing cultures are distinguished into POF$^+$ and POF$^-$, respectively [100]. The genes responsible for ferulic acid decarboxylation are *PAD1* and *FDC1* encoding a flavin prenyltransferase and ferulic acid decarboxylase, respectively. Pad1 contributes to the formation of a flavin-derived cofactor required by Fdc1 for decarboxylating the precursor ferulic acid [101]. Therefore, both genes are required to assure the POF$^+$ phenotype. In a seminal study, Gallone et al. (2016) proved that loss-of-function mutations of *PAD1*, *FDC1*, or both genes are widespread in the genomes of wine, sake, and beer yeasts, while yeasts from the natural environment maintain Pad1 and Fdc1 functional [3]. Ferulic acid decarboxylation is considered a detoxification system which assures yeast survival in the plant-derived environment [102]; thus, loss-of-function mutations of *PAD1* and *FDC1* genes in human-related yeast cultures are a signal of domestication which has selected for the POF$^-$ phenotype [103].

In contrast to Gallone et al. [3], another study reported that the brewing yeast TUM 507 has unfunctional *PAD1*/*FDC1* genes and retains POF$^+$ phenotype, whereas TUM 380 yeast is POF$^-$ despite functional *PAD1* and *FDC1* genes [4]. Guinness yeasts also retain the POF$^+$ phenotype without functional *PAD1* and *FDC1* genes [104]. This evidence suggests that not yet identified enzymes can affect POF production other than Pad1 and Fdc1. Interestingly,

the threshold concentration of 4VG is higher in Guinness stout compared to other beer styles and the mashing regime does not favor ferulic acid release [28], assuring that the POF$^+$ phenotype has not been counter-selected over time in Guinness yeasts.

## 4. New Insights on Strategies to Enhance Yeast Aroma Profile

In recent years, ample research has been focused on applying wild (especially non-*Saccharomyces*) yeasts in beer production. Common characteristics of wild yeast strains include the simultaneous high production of fruity and floral aroma compounds and low ethanol production. On the other hand, CRISPR-Cas technology and other engineering tools allow for the introduction of heterologous metabolic pathways into *S. cerevisiae*, enhancing the yeast contribution to the "floral" and "hoppy" perception of beer. Pros and cons about bioprospecting strategies and synthetic biology approaches are critically discussed, considering the most recent advances in these fields.

### 4.1. S. cerevisiae vs. Nonconventional Yeasts and the Rising of Synthetic Consortia

Brewing yeasts exhibited unique genomic signals of domestication which make them the most powerful cell factories for obtaining lager and ale beer. In addition to the above mentioned POF$^-$ phenotype, another well-documented domestication trait in yeast beer is maltotriose fermentation, a capability linked to the *AGT1* genes in several ale strains [3,103]. Bioprospecting for novel brewing starter cultures entails the usage of both *S. cerevisiae* from alternative niches compared to the brewing environment, as well as *Saccharomyces* non-*cerevisiae* and non-*Saccharomyces* yeasts, as extensively reviewed by [105]. *S. cerevisiae* from sourdough and other fermented food have proved to have good brewing aptitude in terms of fermentative performance, higher alcohols, and esters acetate profile, but they generally exhibit the POF$^+$ phenotype which limits their application only to a few beer specialties [106].

Among non-*Saccharomyces* yeasts, the most promising brewing species are *Torulaspora delbrueckii*, *Hanseniaspora uvarum*, and *Brettanomyces bruxellensis*. They produce lower ethanol and higher concentrations of "fruity" and "floral" compounds than *S. cerevisiae* and have been proposed for producing beers with low ethanol content (0.5–1.2% *v/v*).

*T. delbrueckii* can ferment medium- and high-gravity worts, producing ethanol content ranging from 0.8 to 4%(*v/v*). Some strains can transform hop aroma terpenoids thanks to β-lyase activity, influencing the aroma profile of the final beer [10]. Tataridis et al. [107] used *T. delbrueckii* strains to produce "wheat" style beers and showed that this species consumes maltose slower than the *S. cerevisiae* commercial starter strain used as a control, producing more intensity and complexity in the product (high production of higher alcohols and esters). Gibson et al. [108] identified *T. delbrueckii* as a high producer of 3-methylbutanol (solvent-like flavor) and disregarded this species as a brewing culture, while Canonico et al. [109] proposed *T. delbrueckii* as a brewing starter for producing beers with an aromatic and pleasant taste.

*B. bruxellensis* is known for a horsey flavor and high acidity and it is involved in mixed fermentations of lambic and gueuze beers [110]. It is also known as the main wine and beer spoilage yeast due to its strong tolerance to stressors and the ability to consume various sugars, including dextrins. [10]. This species can decarboxylate ferulic acid into 4VG and convert other hydroxycinnamic acids into volatile phenols as 4-ethylguaiacol (phenolic aroma and sweet) and 4-ethylphenol (phenolic aroma and astringent) [111,112]. Despite these negative traits, when applied appropriately, *Brettanomyces* spp. strains can contribute to exotic flavors (such as pineapple, mango, pear, and grape) due to the high production of ethyl esters [113].

*H. uvarum* has been employed in sequential combination with *S. cerevisiae*, providing low-alcohol beer with a floral aroma thanks to a high ester concentration [114]. The ability of *S. cerevisiae*—*H. uvarum* mixed fermentations to enhance floral attributes has been proven in different wines as well [115].

Additionally, two patents exist for producing nonalcoholic beer beverages (<0.5% *v/v*) by using *Saccharomycodes ludwigii*, a species described as unable to ferment maltose and maltotriose [116]. Even if *Zygosaccharomyces rouxii* is highly proficient at producing ethanol, it is important for brewers to exercise caution when using this species due to the significant release of diacetyl and acetaldehyde [117,118]. For Belgian-style pale ales, *Pichia kudriavzevii* is a particularly promising species, as it produces a rich variety of aromas, including fruity and wine-like notes [119].

In wine making, multi-species and multi-strain yeast "blends" are prepackaged in optimal ratios for enhancing wine fermentation performance and sensory profile. Taking example from these wine cultures, conventional and nonconventional brewing cultures can be used under different inoculation regimes, such as sequential inoculation, coculturing etc., to modulate the resulting flavor outcomes. More recently, several researchers have investigated natural microbial consortia to unravel how microbe-to-microbe interactions can impact the unique flavor complexity associated with spontaneous fermentation, as occurs in specialty beers like lambic beer [17]. This body of knowledge is useful for the control and prediction of spontaneous fermentation, but also with respect to the in vitro design of synthetic consortia suitable to mimic the natural ones under controlled and standardized conditions [120]. Artificial consortia or cocultures are emerging as promising solutions to manage several bioprocesses, such as intestinal probiotic modulation, waste valorization, and pollution bioremediation [121]. They are carefully constructed and refined to optimize bioproduction and consist primarily of two distinct species or even varying strains from the same species, deliberately tailored and enhanced for their intended uses [122]. We can speculate that the usage of synthetic consortia will find application in the brewing sector as well.

### 4.2. Bioprospecting vs. Metabolic Engineering

Yeast biodiversity is a promising source for novel brewing cultures, but the probability of finding suitable candidates among natural variants could be very low when the pattern of desirable phenotypes is highly complex [18]. Whole-genome engineering approaches, including the conventional GMO-free methods (random mutagenesis, hybridization [123,124], and adaptive evolution [125]), as well as the novel GMO approaches (i.e., yeast oligo-mediated genome engineering (YOGE) [126] and eukaryotic multiplex automated genomic engineering (eMAGE) [127]), can circumvent this challenge. These methods expand, via GMO and GMO-free techniques, the available natural diversity and create libraries containing >$10^6$ genetic variants from which strains with superior phenotypes can be isolated. These approaches are scalable when the desired phenotypes are fitness-related, as occurs for tolerance to different stressors (i.e., ethanol and high/low temperature) or for growth on novel substrates. In the case of flavor-active compounds, their biosynthesis does not confer any selective advantage in the presence of a stressor. Consequently, expensive and time-consuming analytic techniques are required to identify new flavor producing variants. In recent years, several high-throughput screening (HTS) devices, based on FACS, microfluidics, and biosensors, have been developed to overcome this caveat, as extensively reviewed by [128]. However, high costs still hinder HTS usage in most research laboratories. In a few cases, the release of sensorial active molecules can be associated with selective growth advantages/disadvantages. For instance, *S. eubayanus* POF⁻ mutants obtained via UV mutagenesis of wild type POF⁺ strains were successfully selected based on their sensitivity towards cinnamic acid and subsequently exploited in a de novo hybridization assay with POF⁻ *S. cerevisiae* ale strains to create synthetic lager brewing yeasts with diverse aroma profiles [129].

As an alternative to the whole-genome approaches described above, metabolic engineering entails bottom-up, gene-targeted methods which have the common goal of improving three essential parameters of target products: titer, rate, and yield (TYR) [130]. As any other engineering discipline, metabolic engineering works according to the Design–Build–Test–Learn cycle and can be successful for three main reasons: (1) advances in

machine learning and computational approaches made the design of genetic manipulations rational and the resulting outputs predicable; (2) advances in synthetic biology have reduced the cost for DNA synthesis and, simultaneously, made DNA assembly easy to perform [131–133]; and (3) new techniques like CRISPR-Cas9 allowed for multiplexing and marker-free genetic manipulation [126,134].

When pathways and genes are known, metabolic engineering improves TYR of natural molecules or introduces novel metabolic pathways into a yeast strain which generally exhibits high fitness and robustness under industrial conditions. Indeed, these latter traits are governed by multiple, complex, and highly interconnected genes and are difficult to modulate via gene-targeted engineering [135]. This is one of the reasons why most attempts for engineering yeast cells involved *S. cerevisiae* and its relatives, such as *S. pastorianus* lager strains [136] and *S. eubayanus* [137]. In addition, *S. cerevisiae* is prone to genetic manipulations thanks to a homologous recombination (HR) machine which repairs chromosomal double-strand breaks (DSB) efficiently. Thanks to this effective HR machine, CRISPR-Cas9 can be more effective in *S. cerevisiae* than in other nonconventional yeasts when it used to introduce single-point mutation into a target gene [138]. For instance, CRISPR-Cas9 has been successfully used to introduce an allelic variant of the *MDS3* gene responsible for high isoamyl acetate production into the tetraploid lager yeast JT28325 [70]. As result, the lager strain harboring the new *MDS3* allelic variant produced beer with significantly higher banana flavor than the wild-type strain. Furthermore, a nonsense mutation in the *FDC1* gene has been introduced via CRISPR-Cas9 in *S. cerevisiae* and *S. eubayanus*, providing POF⁻ strains suitable for in vitro interspecific hybrids construction [137]. Krogerus et al. [139] used CRISPR-Cas9 to increase hybridization yield. In detail, one *MAT* locus was deleted in β-lyase-positive *MAT*a/*MAT*α diploid parental strains via CRISPR-Cas9. The resulting derivatives were forced to mate with a partner with the opposite mating type, giving hybrids with strong β-lyase activity at a high frequency.

The most recent attempts to use brewing yeasts as hosts for heterologous pathway expression aimed at enhancing the release of thiols and monoterpene alcohol production, respectively. Expression of *E. coli* tryptophanase enzyme tnaA with a potent, high β-lyase activity in yeast resulted in a 25-fold increase in released thiol levels from cysteine-bound precursors during must fermentation [84]. Recently, the heterologous expression of *E. coli TnaA* has been coupled with *ATF1* and *ATF2* gene overexpression to increase both the production of 3-sulfanylhexan-1-ol and its ester 3-sulfanylhexyl acetate [140].

Concerning monoterpene alcohols, basic scaffolds containing 10 carbon atoms are produced in plants by the conversion of geranyl diphosphate (GPP) via terpene synthases, such as geraniol synthase (GES) and linalool synthase (LIS). GPP is formed via condensation of DMAPP with its isomer IPP. As reported above, the main target for engineering monoterpene alcohols in yeast is the MVA pathway, which involves seven enzymes to produce DMAPP and IPP. The rate-limiting enzymes are 3-hydroxy-3-methylglutaryl-coenzyme A reductase (HMGR), Hmg1, and IPP isomerase Idi1 [141] (Figure 5). In yeast, GPP can be further condensed with another IPP resulting in farnesyl diphosphate (FPP). FPP is the substrate for the biosynthesis of sesquiterpenes and diterpenes, can dimerize to squalene, and serves as the base molecule in the farnesylation of numerous proteins (Figure 5). In yeast and animals, FPP synthesis is catalyzed by a single enzyme, Erg20, that supports the sequential 1′-4 coupling of IPP with DMAPP, resulting in GPP. *ERG20* gene deletion is lethal, so increased flows towards GPP instead of FPP can be obtained via the overexpression of a gene variant encoding a FPP synthase with low processivity [142]. Taking advances from this notion, Derby et al. [14] produced beer with high titers of geraniol and linalool by combining overexpression of a FPP synthase at low processivity to the heterologous expression of plant-derived monoterpene synthases. The overexpression of a dysregulated form of yeast HMGR lacking the regulatory domain [143] was also carried out to further enhance the carbon flux towards the MVA pathway (Figure 5).

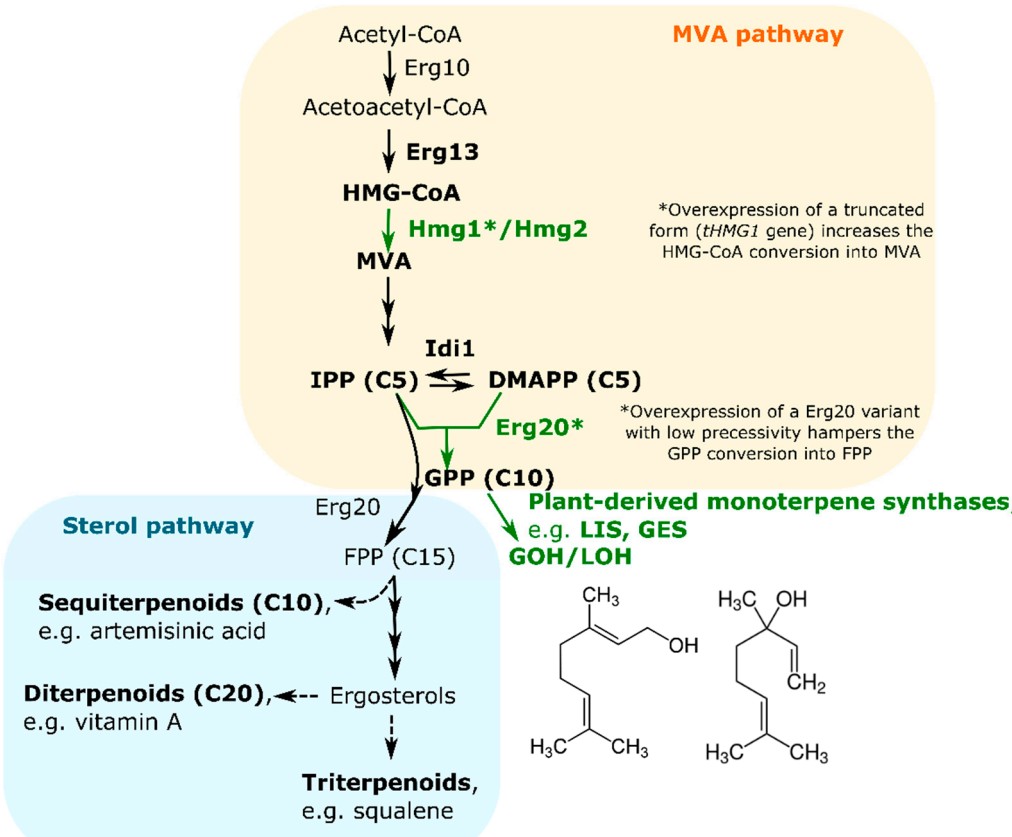

**Figure 5.** Engineering monoterpene alcohols production in yeast. Upstream MVA and downstream sterol biosynthesis pathways are shown. Monoterpene alcohols synthesis is based on its native meval-onate (MVA) pathway, which is linked to the central carbon metabolism via acetyl-CoA. The figure was modified from [14] by adding key intermediates and enzymes and charting the sterol pathway. Genetically modified and/or overexpressed enzymes and ectopically expressed genes are indicated in green. Dotted arrows indicate omitted reactions. Acetyl-CoA, acetyl coenzyme A; HMG-CoA, 3-hydroxy-3-methylglutaryl-CoA; *tHMG1*, truncated HMG-CoA reductase gene; IPP, isopentenyl pyrophosphate; DMAPP, dimethylallyl pyrophosphate; Erg10, acetyl-CoA C-acetyltransferase; Erg13, 3-hydroxy-3-methylglutaryl-CoA synthase; Idi1, isopentenyl-diphosphate δ-isomerase gene; Erg20, farnesyl diphosphate synthetase; FPP, farnesyl diphosphate; LIS, linalool synthase; GES: geraniol synthase; LOH, linalool; GOH: geraniol.

## 5. Conclusions and Future Perspectives

Brewing-yeast research is living in an exciting era. Diversification of the aroma profile encourages the search for novel flavor-active brewing yeasts suitable to give beer a specific character. On the other hand, the brewing industry is reaching for novel solutions to reduce the environmental impact of the production process. The maps now available of genetic markers underpinning flavor-molecule metabolism empowered the marker-assisted selection and design of flavor-active yeast cultures to achieve both goals. The introduction of novel wild yeasts into the brewing production appears useful to renew the yeast cultures portfolio and to provide beers with attractive fruit and floral aromas. However, wild yeasts lack the domestication traits necessary for industrial-scale beer production, such as the capability to ferment maltotriose and the POF$^-$ phenotype. Genetic manipulations are, therefore, required to improve the exploitability of these cultures, but, unfortunately, several consumer concerns still arise. Consumers worldwide are displaying misconceptions and even unfamiliarity with GMO food products. The reluctance to consume beer produced with genetically modified yeast cultures also hampers the feasibility of novel whole-genome engineering methods and synthetic toolkits beyond the laboratory edges.

So, processes generating genotype diversity in nature, such as hybridization and random mutagenesis, could be attractive solutions. If implemented with automatic HTS devices and marker-assisted selection, these methods can dramatically improve the search for novel flavor-active cultures. Furthermore, our knowledge on microbial communities has been increasing in last decade, making a paradigm shift from "axenic culture" to synthetic communities possible. Synthetic communities and yeast blends could be promising solutions for producing beers with a tailored flavor profile as they can mimics the intricate microbial interactions generating beer aroma complexity in spontaneous and/or semi-spontaneous (back-slopping) brewing processes.

**Author Contributions:** L.S. and C.N.: Collection of the information, planning, and drafting of the manuscript; C.N.: Data curation and formatting of the manuscript; L.S.: Conceptualization, analysis, and acquisition of funding. L.S. and C.N.: review editing. All authors have read and agreed to the published version of the manuscript.

**Funding:** The authors acknowledge the support of NBFC to the University of Modena and Reggio Emilia, funded by the Italian Ministry of University and Research, PNRR, Missione 4 Componente 2, "Dalla ricerca all'impresa", Investimento 1.4, Project CN00000033. C.N. was partially supported by a grant from AEB Spa (Brescia, Italy).

**Institutional Review Board Statement:** Not applicabe.

**Informed Consent Statement:** Not applicable.

**Data Availability Statement:** Not applicable.

**Conflicts of Interest:** The funders had no role in the design of this study; in the collection, analyses, or interpretation of the literature; in the writing of the manuscript; or in the decision to publish the review.

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
