# Peer review of "Yeast Bioflavoring in Beer: Complexity Decoded and Built up Again"

_fermentation, doi:10.3390/fermentation10040183_

Round 1

Reviewer 1 Report

Comments and Suggestions for Authors

This review paper describes how yeast serves as a potent bioflavoring agent in beer production, offering complexity and character. It also covers how genetic advancements have unveiled the role of various genes and mutations in yeast bioflavoring, fostering strategies like bioprospecting and metabolic engineering to enhance flavor diversity and sustainability for beer brewers in the face of climate change challenges.

This is a rapidly evolving field and I would have liked to see more recent references cited. The manuscript offers a sub-standard review of the literature, reciting many already known and established facts. The manuscript could benefit from relating those many topics specifically to beer brewing and flavour generation. Additionally, the manuscript reads as a early first draft that requires significant review time to make it readable for the scientific community.

Specific comments

Table 1: Did you determine these to be the main ones based on aroma threshold or something else? Please clarify

Line 49: “The main biotransformation…” please correct sentence structure and grammar.

Line 57: reference please

Line 61: yeast has always been at the forefront of brewing. I am unclear as to the value of the sentence. Please rewrite.

Line 106: Explain why is this the most important pathway in the context of beer brewing?

Line 108 : “In beer wort…” sentence structure and grammar needs fixing.

Figure 5 caption: you have cited reference 14 (Industrial brewing yeast engineered for the production of primary flavor determinants in hopped beer) but have missed some key items from their figure. I suggest adding an explanation as to why you chose to adjust their figure.

Conclusions and future perspectives

Line 530 – 533 “While introducing novel wild yeasts into industrial applications appears useful to renew yeast cultures portfolio, it is also clear that wild yeasts lack of domestication signals necessary for the industrial scale beer production and require manipulations which still arise several consumer concerns.” The sentence is confusing and seems to be joining several topics together. (lack of domestication pressure, adaptation to specific industrial environments, consume views on genetic engineering or lack of industry understanding?) Please rewrite.  

Line 533 - 534 “The same hurdles hinder the broad application of novel whole-genome engineering and synthetic toolkits beyond the laboratory edges” Exactly what hurdles is this referring to? Consumer concerns or the domestication of wild yeasts or both? Please rewrite.

Line 535: “So, looking towards the nature helps us solving this caveat” Grammer and sentence structure needs fixing.

Line 538 – 541: “On the other ….” I am unclear as to what the conclusion here is pertaining to the manuscript. It seems off topic to the rest of the manuscript and it only relates to specific brewing processes. Please amend so it is more appropriate to the manuscript topic.   

Comments on the Quality of English Language

Already commented. 

Author Response

Point-by-point Response to issues arisen from Reviewer 1

This review paper describes how yeast serves as a potent bioflavoring agent in beer production, offering complexity and character. It also covers how genetic advancements have unveiled the role of various genes and mutations in yeast bioflavoring, fostering strategies like bioprospecting and metabolic engineering to enhance flavor diversity and sustainability for beer brewers in the face of climate change challenges.

This is a rapidly evolving field and I would have liked to see more recent references cited. The manuscript offers a sub-standard review of the literature, reciting many already known and established facts. The manuscript could benefit from relating those many topics specifically to beer brewing and flavour generation. Additionally, the manuscript reads as a early first draft that requires significant review time to make it readable for the scientific community.

R: We extensively modified the manuscript to increase its quality. We also added some more up-to-date references. In any case, we believed that an overview of the state-of-art can help the reader in understanding the most specific and highly complex topics.

Specific comments

Table 1: Did you determine these to be the main ones based on aroma threshold or something else? Please clarify

R: We selected the most significant and studied compounds based on the literature. In any case, we modified the Table to .

Line 49: “The main biotransformation…” please correct sentence structure and grammar.

R: We apologize for the mistake. The statement has been corrected.

Line 57: reference please

R: Done

Line 61: yeast has always been at the forefront of brewing. I am unclear as to the value of the sentence. Please rewrite.

R: Our intention has been to write that selection of novel yeasts is critical to achieve the beer diversification and this topic is on the forefront of brewing innovation. We modified the statement according to Reviewer’s suggestion.

Line 106: Explain why is this the most important pathway in the context of beer brewing?

R: We agree that this pathway is critical for the production of vicinal diketones and detailed better this point.

Line 108 : “In beer wort…” sentence structure and grammar needs fixing.

  1. We modified the statement.

Figure 5 caption: you have cited reference 14 (Industrial brewing yeast engineered for the production of primary flavor determinants in hopped beer) but have missed some key items from their figure. I suggest adding an explanation as to why you chose to adjust their figure.

R: We explained why we modified the Denby’s Figure. We also added the missing key items to the caption.

Conclusions and future perspectives

     Line 530 – 533 “While introducing novel wild yeasts into industrial applications appears useful to renew yeast cultures portfolio, it is also clear that wild yeasts lack of domestication signals necessary for the industrial scale beer production and require manipulations which still arise several consumer concerns.” The sentence is confusing and seems to be joining several topics together. (lack of domestication pressure, adaptation to specific industrial environments, consume views on genetic engineering or lack of industry understanding?) Please rewrite. 

R: We apologize for this and modified the sentence. Different topics were divided into distinct sentences.

Line 533 - 534 “The same hurdles hinder the broad application of novel whole-genome engineering and synthetic toolkits beyond the laboratory edges” Exactly what hurdles is this referring to? Consumer concerns or the domestication of wild yeasts or both? Please rewrite.

R: We apologize for this and rewrote the statement to explain pro and contra of all the strategies adopted to manipulate flavor-active yeasts.

Line 535: “So, looking towards the nature helps us solving this caveat” Grammer and sentence structure needs fixing.

Line 538 – 541: “On the other ….” I am unclear as to what the conclusion here is pertaining to the manuscript. It seems off topic to the rest of the manuscript and it only relates to specific brewing processes. Please amend so it is more appropriate to the manuscript topic.  

R: We modified the conclusions to focus it better on the first part of the manuscript too.

Comments on the Quality of English Language

Already commented. 

R: We extensively reviewed the English language and hope that the manuscript can now fulfill the standards of quality.

Reviewer 2 Report

Comments and Suggestions for Authors

Review : Nasuti et al. Yeast bioflavoring

This is a good and helpful review article. I recommend publication after the indicated corrections. Editorial corrections are included as insertions, deletions, and comments to the submitted file. Substantive issues are also mentioned in the list below.

Line 29 Too broad. Different brewing steps come to the fore in different styles. In a hop-forward style, boiling is most influential. In some lager styles, conditioning is most important.

Line 31: Most aroma-active compounds come from hops.

Line 32: A “yeast cell factory” would be a system that manufactures yeast cells. Explain what you mean.

Line 37: This statement is incorrect. In most lagers, carbon dioxide, hop bitterness, and ethanol are present above their thresholds.

Line 52: The relative cost of malt and hops varies depending on the recipe and on the prevailing prices, but usually malt is the major raw material cost. The requirements for water, energy, and land to produce hops for a barrel of beer are barely significant compared to those for malt, considering that the hop requirement is about 3% of the malt requirement,

Line 87: IAA may be the most important fusel alcohol for beer flavor, but it is not as important as CO2 or iso-alpha acids.

Figure 1: In the oxidation of the aldehyde to acid, NADH and NAD+ should be reversed: NAD+ +2H --> NADH + H+. Caption for this figure should include only material describing the illustration.

Lines 157-158. Be clear about what is the numerator and what is the denominator in the ratio.

Line 186: Mentions fusel Acyl-CoA and fatty acid Acyl=l-CoA. Explain the distinction.

Line 206: Explain how beer fermentation is semi-aerobic.

Line 223: Hop oil does not contribute bitterness, only aroma.

Comments on the Quality of English Language

The use of the comma (,) and of definite and indefinite articles ("the" and "a") can be tricky even for native speakers.

Don't use reference numbers as text.
No: This was shown by [86].
Yes: This was shown by King et al. [86]. 

Don't use mathematical operators or other symbols as text.
No: The price of a kilogram of hops is > than that of malt.
Yes: The price of a kilogram of hops is higher than that of malt.

Only use quotation marks for direct quotations.
No: "banana" flavor.
Yes: banana flavor.
Yes: Priestly called oxygen "dephogisticated air".

Author Response

Point-by-point Response to issues arisen from Reviewer 2

This is a good and helpful review article. I recommend publication after the indicated corrections. Editorial corrections are included as insertions, deletions, and comments to the submitted file. Substantive issues are also mentioned in the list below.

Line 29 Too broad. Different brewing steps come to the fore in different styles. In a hop-forward style, boiling is most influential. In some lager styles, conditioning is most important.

R: We modified the statement according to reviewer’s concerns.

Line 31: Most aroma-active compounds come from hops.

R: We agree that hop is very important for beer quality, but in this work, we focused on the yeast contribution. We tried to better explain this point.

Line 32: A “yeast cell factory” would be a system that manufactures yeast cells. Explain what you mean.

R: We mean “yeasts used as cell factories” to produce flavor. In any case, we used “culture”.

Line 37: This statement is incorrect. In most lagers, carbon dioxide, hop bitterness, and ethanol are present above their thresholds.

R: We explained that other flavor molecules are present other than carbon dioxide, hop bitterness, and ethanol.

Line 52: The relative cost of malt and hops varies depending on the recipe and on the prevailing prices, but usually malt is the major raw material cost. The requirements for water, energy, and land to produce hops for a barrel of beer are barely significant compared to those for malt, considering that the hop requirement is about 3% of the malt requirement,

R: We added “malt” to the statement.

Line 87: IAA may be the most important fusel alcohol for beer flavor, but it is not as important as CO2 or iso-alpha acids.

R: We recognized that IAA are important for beer flavor but here we focused the discussion on yeast-derived flavor molecules.

Figure 1: In the oxidation of the aldehyde to acid, NADH and NAD+ should be reversed: NAD+ +2H --> NADH + H+. Caption for this figure should include only material describing the illustration.

R: Thanks, we modified both caption and figure.

Lines 157-158. Be clear about what is the numerator and what is the denominator in the ratio.

R: We modified the statement.

Line 186: Mentions fusel Acyl-CoA and fatty acid Acyl=l-CoA. Explain the distinction.

R: We slightly modified the statement to better explain the difference. Definition of fucel Acyl-CoA and fatty acyl-CoA were from Bisson & Karpel 2010 (Bisson, L.F.; Karpel, J.E. Genetics of yeast impacting wine quality. Annu. Rev. Food Sci. Technol. 2010, 1, 139–162

Line 206: Explain how beer fermentation is semi-aerobic.

R: We delated this word.

Line 223: Hop oil does not contribute bitterness, only aroma.

R: sorry for the mistake, we modified the statement accordingly.

Comments on the Quality of English Language

The use of the comma (,) and of definite and indefinite articles ("the" and "a") can be tricky even for native speakers.

R: We strongly revised the manuscript to improve the English language quality.

Don't use reference numbers as text.
No: This was shown by [86].
Yes: This was shown by King et al. [86]. 

R: OK

Don't use mathematical operators or other symbols as text.
No: The price of a kilogram of hops is > than that of malt.
Yes: The price of a kilogram of hops is higher than that of malt.

R: Done.

Only use quotation marks for direct quotations.
No: "banana" flavor.
Yes: banana flavor.

R: Done
Yes: Priestly called oxygen "dephogisticated air"

R: Sorry, I did not understand this last correction.

Reviewer 3 Report

Comments and Suggestions for Authors

Just one comment, at the beginning of the review you start mentioning amounts and thresholds of the different aromatic molecules. That information does not appear or is revisited after the first half of the paper. I would include the information to increase the importance of this research. 

Author Response

Point-by-point Response to issues arisen from Reviewer 3

R: We added the required information.

Round 2

Reviewer 1 Report

Comments and Suggestions for Authors

Still the occasional sentence structure and grammar mistakes that need fixing but I am satisfied with the manuscript.

Comments on the Quality of English Language

Some words have no space between them. Occasional sentence structure error and gramm